# The COVID-19 Vaccine and Pregnant Minority Women in the US: Implications for Improving Vaccine Confidence and Uptake

**DOI:** 10.3390/vaccines10122122

**Published:** 2022-12-12

**Authors:** Donald J. Alcendor, Patricia Matthews-Juarez, Duane Smoot, James E. K. Hildreth, Mohammad Tabatabai, Derek Wilus, Katherine Y. Brown, Paul D. Juarez

**Affiliations:** 1Department of Microbiology, Immunology and Physiology, Center for AIDS Health Disparities Research, School of Medicine, Meharry Medical College, 1005 Dr. D.B. Todd Jr. Blvd., Nashville, TN 37208, USA; 2Department of Family & Community Medicine, Meharry Medical College, 1005 D.B. Todd Jr. Blvd., Nashville, TN 37208, USA; 3Department of Internal Medicine, School of Medicine, Meharry Medical College, 1005 D.B. Todd Jr. Blvd., Nashville, TN 37208, USA; 4School of Graduate Studies and Research, Meharry Medical College, 1005 D.B. Todd Jr. Blvd., Nashville, TN 37208, USA

**Keywords:** COVID-19, vaccinations, influenza, Tdap, pregnancy, minorities, women, vaccine hesitancy

## Abstract

The American College of Obstetricians and Gynecologists (AGOG) recommends the FDA-approved Pfizer and Moderna mRNA COVID-19 vaccines and boosters for all eligible pregnant women in the US. However, COVID-19 vaccine confidence and uptake among pregnant minority women have been poor. While the underlying reasons are unclear, they are likely to be associated with myths and misinformation about the vaccines. Direct and indirect factors that deter minority mothers in the US from receiving the mRNA COVID-19 vaccines require further investigation. Here, we examine the historical perspectives on vaccinations during pregnancy. We will examine the following aspects: (1) the influenza and tetanus toxoid, reduced diphtheria toxoid, and acellular pertussis (Tdap) vaccinations during pregnancy; (2) the exclusion of pregnant and lactating women from COVID-19 vaccine trials; (3) COVID-19 vaccine safety during pregnancy, obstetric complications associated with symptomatic COVID-19 during pregnancy, COVID-19 vaccine hesitancy among pregnant minority women, and racial disparities experienced by pregnant minority women due to the COVID-19 pandemic as well as their potential impact on pregnancy care; and (4) strategies to improve COVID-19 vaccine confidence and uptake among pregnant minority women in the US. COVID-19 vaccine hesitancy among minority mothers can be mitigated by community engagement efforts that focus on COVID-19 vaccine education, awareness campaigns by trusted entities, and COVID-19-appropriate perinatal counseling aimed to improve COVID-19 vaccine confidence and uptake.

## 1. Introduction

Pregnancy has been shown to be a risk factor for severe COVID-19 and its associated mortality [1,2,3,4]. The rate of hospital admission to intensive care for pregnant women with COVID-19 is three times greater than that for non-pregnant women; the former also experience a 25% greater risk of death [5]. However, pregnant women are routinely excluded from vaccine trials mainly due to liability and safety concerns for the mother and baby. Pregnant and lactating women were excluded from the COVID-19 vaccine trials despite their tendency to suffer from more severe complications when stricken with COVID-19 [6]. Unvaccinated pregnant women stricken with COVID-19 are more likely than vaccinated pregnant women to experience symptoms and complications as well as require hospitalization, admission to the intensive care unit (ICU), and mechanical ventilation [7,8,9]. They are also more likely to encounter preterm birth and experience neonatal complications compared to non-gravid females without COVID-19 [10,11]. The Centers for Disease Control and Prevention (CDC), American College of Obstetricians and Gynecologists (ACOG), and Society for Maternal-Fetal Medicine (SMFM) recommend the FDA-approved Pfizer and Moderna mRNA COVID-19 vaccines and boosters for all eligible pregnant and lactating women in the US [12,13,14]. Despite strong recommendations from experts in women’s health regarding the effectiveness of the COVID-19 vaccines, poor COVID-19 vaccine confidence and uptake persist among all women, especially minority women [15,16]. When pregnant women vaccinated against COVID-19 are infected, they are less likely to experience symptoms, require hospitalization and mechanical ventilation, and die from COVID-19. Transplacental transfer of maternal antibodies from mothers vaccinated against COVID-19 may provide protection against neonatal infection [17,18,19,20]. In a study by Prahl et al. of a cohort of 20 women during late pregnancy, the authors found no evidence of mRNA vaccine products in maternal blood, placenta tissue, or cord blood at delivery [21]. Interestingly, they observed a time-dependent transfer of IgG and neutralizing antibodies to the neonates that persisted during early infancy [21]. Moreover, they observed sequence-specific binding of the antibodies to the spike protein epitope in the infants. Therefore, the authors concluded that timely immunization of pregnant women against COVID-19 is essential to ensure efficient transplacental transfer of protective antibodies during early infancy [21].

A study by Halasa et al. showed maternal completion of the primary two-dose mRNA COVID-19 vaccination series during pregnancy to protect infants less than six months of age from COVID-19-associated hospitalization (61%; 95% CI = 31% to 78%) [22]. The authors also showed the completion of the primary COVID-19 vaccine series in early and late pregnancy to be at 32% (95% CI = −43% to 68%) and 80% (95% CI = 55% to 91%), respectively [22]. Despite the known benefits of the COVID-19 vaccines for expectant mothers and their babies, vaccine uptake continues to be the lowest among African American and Hispanic women in the US [23].

Here, we explore the types of vaccines recommended during pregnancy, the safety and benefits of the COVID-19 vaccines during pregnancy, and the racial and ethnic disparities impacting pregnant minority women that could disproportionately expose them to COVID-19 and inequities in vaccine access.

## 2. Historical Perspectives on Vaccination during Pregnancy

Vaccination is a vital component of pregnancy care that has greatly improved maternal and perinatal outcomes over the years [24]. Pregnant women are at a greater risk of certain infections due to a pregnancy-induced shift from cell-mediated immunity (Th1 response) to humoral immunity (Th2 response) [25]. This adaptation protects a semi-allogeneic fetus from immunologic rejection by the mother. However, it also puts the mother at a greater risk of certain infections. In addition, the cardiopulmonary adaptations during pregnancy to accommodate the developing fetus also increase the mother’s susceptibility to life-threatening respiratory tract infections [25]. Observational and prospective studies have shown that the influenza and tetanus vaccines can reduce maternal and infant disease burden [26,27,28,29,30]. Maternal immunization can close the immunity gap for children that are too young to be immunized and will likely be expanded in the future to include other vaccines such as those against group B streptococcus and respiratory syncytial virus (RSV) [31]. Maternal immunization is also warranted for emergency travel and under special circumstances to protect mothers and babies against severe and life-threatening diseases. Women who are vaccinated against specific infections during pregnancy will develop maternal antibodies that can cross the placental barrier to protect their babies [32].

## 3. The Historical Impact of the Influenza Vaccine on Pregnancy Care

Primary influenza virus infection can lead to considerable morbidity and mortality in pregnant women. The Spanish flu pandemic of 1918 resulted in approximately 50–100 million deaths globally, with higher incidence among pregnant women. In a study by Woolston et al., the authors observed a mortality rate of 45% among pregnant women admitted with influenza complicated by pneumonia, compared to 32% among 2053 non-pregnant women admitted with the same illness [33]. A case report by medical practitioners in the US reported a fatality rate of 27% among 1350 pregnant women with influenza, and mortality was observed mainly among cases complicated by pneumonia [34]. Although the influenza pandemic of 1957 and 1968 did not result in similar mortality rates, mortality among pregnant women during these pandemics was disproportionately high. Notably, in 1957, influenza was the leading cause of death (~20%) among pregnant women [35]. Approximately 50% of women of reproductive age who died of influenza during the 1957 pandemic were pregnant [35]. The inactivated monovalent influenza A subtype vaccine was developed in the 1930s; the US military tested the vaccine in 1938 [36]. A large-scale trial conducted in the US between 1942 and 1945 led to the development of the bivalent/trivalent vaccines that were licensed for public use [37]. When the disproportionate morbidity and mortality due to influenza continued to persist among pregnant women, in 1960, public health authorities in the US recommended that pregnant women be prioritized to receive the then widely available inactivated trivalent influenza vaccine [38]. However, it was not until 1997 that CDC endorsed the public health authorities’ recommendation that pregnant women be vaccinated against influenza [39]. Although clinical trial data on the efficacy of the influenza vaccine in pregnant women were limited, immunogenicity studies showed pregnant women who had received the vaccine developed immune-protective responses [40,41]. A randomized clinical trial known as the Mother’s Gift Project was conducted in Bangladesh between 2004 and 2005. The trial demonstrated that pregnant women who had received an influenza vaccine were 36% less likely to suffer from respiratory illness with fever compared to pregnant women who had received a pneumococcal vaccine. The study showed a 63% lower risk of laboratory-confirmed influenza among infants less than six months of age whose mothers had been vaccinated [42]. A randomized placebo control trial conducted in Nepal in 2011 and 2013 confirmed earlier findings of the maternal and neonatal protective effects of the influenza vaccine [43].

While the maternal and perinatal benefits of the influenza vaccine are well established, influenza vaccine uptake among pregnant women in the US was only 54.5% in April of 2021 [44]. This suggests a need for influenza vaccine education and awareness to address distrust and safety concerns [45]. Pregnant mothers with influenza are at a greater risk of severe maternal morbidity as well as influenza-related maternal and obstetric complications [46]. In a study by Wen et al., the authors found that from 2000–2018 in the US, pregnant women with an influenza diagnosis at delivery experienced an increased risk of severe maternal morbidity compared to those without influenza (2.3% vs. 0.7%; adjusted risk ratio 2.24, 95% CI 2.17–2.31) [47]. Among the 74.7 million delivery-associated hospitalizations during the same period, the rate of severe maternal morbidity was higher among women with an influenza diagnosis than those without influenza (86–410 cases vs. 53–70 cases/10,000 delivery-associated hospitalizations) [47].

The CDC analyzed the survey data concerning the 2020–2021 flu season [48] that were weighted to reflect the pregnancy status at the time of interview, age, race/ethnicity, and geographic distribution of pregnant women within the total U.S. population.

## 4. The Historical Impact of the Tetanus Toxoid, Reduced Diphtheria Toxoid, and Acellular Pertussis (Tdap) Vaccine on Pregnancy Care

The Tdap (Tetanus, Diphtheria, and Pertussis) vaccine protects against tetanus, diphtheria and pertussis and is composed of three separate inactivated bacterial toxins from Clostridium tetani, *Corynebacterium diphtheria, Bordetella pertussis*, respectively, that were developed for pregnant women to provide neonatal protection via passive transfer of protective antibodies to the developing fetus [49]. While the incidence of neonatal tetanus around the world has decreased significantly, many developing countries still experience significant morbidity and mortality due to neonatal tetanus [50]. Neonatal tetanus is usually acquired via contamination of wounds or abrasions by spores from bacterium *Clostridium tetani* that can be found in soil. In developing countries, neonatal tetanus often occurs due to contamination of the umbilical stump resulting from unsanitary birth practices [51]. Infected neonates clinically present with generalized rigidity, muscle spasms, and a loss of the ability to breastfeed [51]. The case fatality rate in the absence of medical intervention approaches 100%, while mortality is between 10 and 60% for infants who receive intensive care [52].

Licensed in 1938, the tetanus toxoid vaccine was distributed widely during World War II and observational studies in 1960 showed multiple doses of the vaccine were required to prevent neonatal tetanus [53]. A double-blind study revealed that while a single dose of the vaccine was not effective at preventing neonatal tetanus, two or three doses were effective [53]. Follow-up studies showed 94% vaccine efficacy in reducing mortality [54]. During the 1980s, neonatal tetanus was responsible for approximately 500,000 neonatal deaths worldwide [55]. A resolution by the World Health Assembly was passed in 1988 to eliminate neonatal tetanus by year 2000 [55]. Among the 1000 live-born infants, there were 6.7 deaths due to neonatal tetanus and the primary focus of the vaccine study was on children, pregnant women and women of childbearing age, as well as on hygienic peripartum care practices. The campaign was highly successful, resulting in a 93% reduction in tetanus-associated neonatal death, and 47 out of the 59 countries were able to achieve elimination status [55]. In 2017, neonatal tetanus mortality fell from an estimated 787,000 to 31,000 cases worldwide [56]. Nonetheless, maternal morbidity and mortality due to tetanus continue to be concerning in some countries.

## 5. The Historical Impact of the Pertussis Vaccine on Pregnancy Care

Pertussis, commonly known as whooping cough, is a highly infectious respiratory disease caused by bacterium *Bordetella pertussis*. Clinical presentations for pertussis include paroxysmal cough, inspiratory whoop, and post-tussive emesis. High mortality rates were common during the 1930s. For instance, there were 36,103 deaths from pertussis among mostly young infants in the US between 1926 and 1930 [57]. With the development of the whole-cell pertussis vaccine in the 1940s, widespread vaccination of pregnant women resulted in a dramatic reduction in the incidence of pertussis, from a peak of 250,000 cases in 1943 to 1010 cases in 1976 [58]. Subsequently, vaccination of pregnant women to impart passive immunity to their babies was supplemented with the inclusion of vaccination against pertussis in the infant scheduled vaccinations in the US, resulting in further reduction in pertussis-related infant mortality [59]. According to the Advisory Committee on Immunization Practices (ACIP), inactivated vaccines can be administered safely during pregnancy [60]. In the US, it is recommended that all pregnant women receive the seasonal inactivated influenza vaccine and the Tdap (trivalent tetanus toxoid, reduced diphtheria toxoid, and acellular pertussis vaccine, the latter recommended during the third trimester) [60]. The influenza and Tdap vaccines can decrease the risk of flu and pertussis, respectively, among pregnant women and their infants. The COVID-19 vaccines are also recommended for all pregnant and lactating women in the US to reduce COVID-19-associated risk of morbidity and mortality in this population [61]. Current vaccine regimens, dosing, and complications for unvaccinated pregnant women are shown in Figure 1.

Currently recommended vaccines for all pregnant and lactating women in the US include the Tdap vaccine, the influenza vaccine, and the COVID-19 vaccines from Pfizer and Moderna. Dosages, as well as perinatal and maternal protections, are listed, along with maternal and perinatal complications for infected mothers who are unvaccinated.

## 6. COVID-19 Vaccination Trials for Pregnant and Lactating Women

Although early clinical trials for the COVID-19 mRNA vaccines demonstrated the vaccines’ safety and effectiveness in adults across various demographics, pregnant and lactating women were excluded from these trials. Women that were pregnant and infected with COVID-19 were shown to face a greater risk of hospitalization and ICU admission as well as be more likely to require mechanical ventilation and experience preterm birth compared to uninfected pregnant women [62,63,64]. Most notably, Pfizer and Moderna systematically excluded pregnant and lactating women from their COVID-19 vaccine trials and thus did not obtain evidence to suggest that the vaccines were teratogenic or being secreted in breast milk. Because mRNA vaccines are quickly degraded after injection into the muscle tissue and their post-immune effects appear to be mild, the rationale for excluding pregnant and lactating women from the clinical trials is unclear [65]. This systematic exclusion put many healthcare professionals who are women and were either pregnant or had just given birth at a greater risk of severe complications due to COVID-19 because of the nature of their professions [66]. Moreover, when healthcare workers are infected, they become a source of nosocomial transmission to their patients and colleagues. In addition, in the absence of clinical trial data, the reason for systematically excluding pregnant and lactating women from the COVID-19 vaccine trials when these two are biologically different from each other is unclear. In the eligibility criteria for clinical trials, pregnant and lactating women are often grouped together even though therapies that are linked to teratogenicity may not affect breast milk and vice versa. Too often, the exclusion of pregnant and lactating women from clinical trials is based solely on observational data from studies on historic drugs rather than data on safety or efficacy [67]. COVID-19 vaccine manufacturers that excluded pregnant and lactating women from their initial clinical trials are Pfizer, Moderna, Johnson & Johnson/Janssen, AstraZeneca, Sinopharm, and Sinovac [68]. Clinical trial exclusion of women contemplating pregnancy could have far-reaching consequences because it reduces the number of women available to participate in clinical trials [68]. It is well-known that pregnant and lactating women are routinely excluded from clinical trials because of safety and liability concerns for the mother and the developing fetus [69,70,71]. However, the exclusion of pregnant and lactating women from COVID-19 vaccine trials can deprive these populations from lifesaving interventions as well as put them at a greater risk of catching the disease and transmitting it to others. One must also consider the benefits of maternal COVID-19 vaccination to the baby due to protective effects of passive immunity from the mother. Therefore, the exclusion of pregnant and lactating women from clinical trials should be reexamined in order to establish new guidelines to ensure that the risk of exposure, infection, and severe clinical outcomes are carefully examined for these populations. Most notably, the National Institute of Allergy and Infectious Diseases (NIAID) has developed guidelines for assessing the safety of pregnant women in clinical trials. In 2018, the FDA responded positively for pregnant women to be included in clinical trials. Even more, the Task Force on Research Specific to Pregnant Women and Lactating Women supported inclusion of pregnant and lactating women in clinical trials unless there was strong scientific evidence to support their exclusion [72].

## 7. The Safety of the mRNA COVID-19 Vaccines during Pregnancy

The ACIP, ACOG, SMFM, and Academy of Breastfeeding Medicine (ABM) recommend COVID-19 vaccination among pregnant and lactating women. Evidence supporting the safety and efficacy of the mRNA COVID-19 vaccines in pregnant and lactating women is strong. A prospective cohort study performed by Gray et al. examined 131 reproductive-age COVID-19 vaccine recipients including 84 pregnant, 31 lactating, and 16 non-pregnant women) [73]. Authors observed robust humoral immunity in both lactating and pregnant women; immune responses were similar to those observed in non-pregnant women. Transfer of protective antibodies to neonates via the placenta and breast milk were also observed [73]. In a systematic review and meta-analysis, Ma et al. examined observational studies on the safety and efficacy of the mRNA COVID-19 vaccines in pregnant women. Their analysis included observational studies performed from the date of EUA vaccine recommendation to 6 December 2021. In all six observational studies included in the analysis, all showed that vaccination prevented pregnant women from SARS-CoV-2 infection (OR = 0.50, 95% CI, 0.35–0.79) and COVID-19-related hospitalization (OR = 0.50, 95% CI, 0.31–0.82) [74]. In addition, the analysis did not reveal any adverse effects of COVID-19 vaccination on pregnancy, fetal, or neonatal outcomes. In a study by Shimabukuro et al., the authors analyzed data from the safety surveillance registries from 14 December 2020, to 28 February 2021 that include information from v-safe and the Vaccine Adverse Event Reporting System (VAERS) for pregnant women who received the mRNA COVID-19 vaccines [75]. The v-safe participants included 35,691 pregnant women who reported headache, fatigue, injection-site pain, and myalgia as the most common reactions after vaccination, which increased in incidence after the second dose. Less than 1% and 8% of the vaccine recipients reported fevers >38 °C after the first and second dose, respectively [75]. There was no difference in the reporting of severe reactions in pregnant and non-pregnant women, although nausea and vomiting were reported more often among pregnant women after the second dose. The authors concluded that the reporting trends were similar between pregnant and non-pregnant vaccine recipients.

## 8. COVID-19 Morbidity and Mortality in Pregnant Minority Women in the US

Pregnant minority women in the US experience greater mortality due to COVID-19-associated complications compared to non-Hispanic Whites [76]. There was a 16.8% increase in overall US mortality in 2020, largely attributed to the COVID-19 pandemic. The National Center for Health Statistics (NCHS) reported an 18.4% increase in maternal mortality in the US (i.e., death during pregnancy or within 42 days of pregnancy between 2019 and 2020 [76]. Increase in maternal death was reported to be 44.4% among Hispanic, 25.7% among non-Hispanic Black, and 6.1% among non-Hispanic White women [77]. Chinn et al. conducted a study on a cohort of 869,079 adult women, including 18,715 women with COVID-19, who underwent childbirth at 499 US medical centers between 1 March 2020, and 28 February 2021 [78]. The authors found that women with COVID-19 experienced an increase in mortality, the need for intubation and ventilation, and ICU admission [78]. The study also found that women who had COVID-19 were more likely to be Black or Hispanic when Blacks and Hispanics make up only 12.4% and 18.7% of the US population according to the 2020 census [78]. A study by Knight et al. examined a cohort of 427 pregnant women admitted to hospital with confirmed SARS-CoV-2 infection between 1 March 2020 and 14 April 2020 [79]. They observed that 233 (56%) of the pregnant women admitted with COVID-19 were black or from other ethnic minority groups, 175 (41%) were aged 35 or older, and 145 (34%) had pre-existing comorbidities 281 (69%) were overweight or obese, [79]. In addition, 266 (62%) women gave birth or had a pregnancy loss; 196 (73%) gave birth at term. Twelve (5%) of 265 infants tested positive for SARS-CoV-2 RNA, six of them within the first 12 h after birth. There were 41 (10%) women admitted to hospital and needing respiratory support, and five women (1%) did not survive. [79]. Unvaccinated pregnant women stricken with COVID-19 face a greater risk of symptomatic disease, hospitalization, ICU admission, and neonatal complications (Figure 2). COVID-19 vaccination in pregnant women can help mitigate these risks (Figure 2).

Shown here are the maternal changes in immune physiology that result in an increased risk of infection by respiratory pathogens (influenza and COVID-19); also shown here are the maternal and perinatal benefits and risks of getting and foregoing COVID-19 vaccination, respectively, prior to infection.

## 9. Obstetric Complications Associated with Symptomatic COVID-19 in Pregnant Women

A multinational cohort study that involved 43 institutions in 18 countries was conducted from March to October 2020 to examine COVID-19-associated maternal and neonatal/perinatal morbidity and mortality. In this study, the researchers observed admitted mothers and their neonates for COVID-19-associated complications until the patients were discharged. A total of 706 pregnant women with a COVID-19 diagnosis and 1424 pregnant women without a COVID-19 diagnosis were enrolled [80]. In this study, the women diagnosed with COVID-19 were at a higher risk of preeclampsia/eclampsia (relative risk [RR], 1.76; 95% CI, 1.27–2.43), severe infection (RR, 3.38; 95% CI, 1.63–7.01), ICU admission (RR, 5.04; 95% CI, 3.13–8.10), and maternal mortality (RR, 22.3; 95% CI, 2.88–172) [80]. Neonatal complications observed included preterm birth (RR, 1.59; 95% CI, 1.30–1.94), severe neonatal morbidity (RR, 2.66; 95% CI, 1.69–4.18), medically indicated preterm birth (RR, 1.97; 95% CI, 1.56–2.51), and severe perinatal morbidity and mortality (RR, 2.14; 95% CI, 1.66–2.75) [80]. In these cases, viral infection of the placenta caused SARS-CoV-2 placentitis, a condition that presents with increased fibrin deposition that typically reaches the level of massive perivillous fibrin deposition, chronic histiocytic intervillositis, and trophoblastic necrosis [81,82,83,84,85]. Therefore, SARS-CoV-2 can cause significant placental damage and infect the fetus, resulting in preterm birth, stillbirth, and neonatal death. In a study by Theiler et al., the authors observed a lower COVID-19 infection rate among vaccinated pregnant women than among unvaccinated pregnant women [86]. In addition, they found no structured pattern of adverse maternal or neonatal clinical outcomes in a cohort of 140 vaccinated pregnant women [87]. Neonatal complications associated with COVID-19 infection in unvaccinated pregnant women include preterm birth, stillbirths, placentitis/trophoblastic necrosis, placental insufficiency, and perinatal death (Figure 3). COVID-19 vaccination in pregnant women can reduce the risk of disease transmission to their babies, fetal infection, preterm birth, stillbirth, and perinatal death (Figure 3). Finally, a study performed in Italy by Di Favio et al., who examined the safety of several COVID-19 vaccines available in Italy from December 2020 to September 2021, showed that the vaccines were considered relatively safe and acceptable [87].

## 10. COVID-19 Vaccine Hesitancy among Pregnant Minority Women

The World Health Organization Strategic Advisory Group of Experts on Immunization (SAGE) has defined vaccine hesitancy as a delay in accepting or refusing immunization regardless of the availability of vaccination services [88]. Pregnant women are generally more reluctant than the general public to accept vaccines due to safety concerns for themselves and their babies. Bhattacharya et al. examined 17 studies on vaccine hesitancy from four continents that included 25,147 pregnant women in total. The authors observed an overall COVID-19 vaccine acceptance prevalence of 49%. The lowest COVID-19 vaccine acceptance prevalence was found in the Americas, with an acceptance prevalence of 45%. Vaccine acceptance was also lower among individuals with less education and who were unemployed, with parity of >1 [89]. Factors that drive vaccine hesitancy can differ based on the vaccine and the target population. Underserved pregnant minority women in the US have been disproportionately impacted by severe COVID-19 disease and COVID-19 vaccine hesitancy [90]. This has resulted in reduced vaccine confidence as well as poor uptake of other vaccines for themselves and their children [91]. The underlying reasons for poor vaccine confidence and vaccine hesitancy among pregnant minority women for the COVID-19 vaccines, as well as for other routine vaccines, require further investigation. In a study by Kiefer et al. that examined COVID-19 vaccine hesitancy among 435 pregnant women, the authors found an average vaccine hesitancy rate of 46% (95% CI, 41–51%) [92]. They observed the highest level of vaccine hesitancy among young non-Hispanic Blacks (52%) with lower education level and public health insurance with parity >1 [92]. They observed that pregnant women in the study who had friends or family who had been vaccinated for COVID-19 also planned to receive other recommended vaccines (influenza and Tdap), perceived that the vaccination would benefit their children and were less likely to be hesitant [92]. Bartarbee et al. conducted a cross-sectional multicenter study from August to December 2020 to examine attitudes toward COVID-19 illness and COVID-19 vaccination among 913 pregnant women [93]. The authors found that while 72% of the women were concerned about becoming sick with COVID-19, only 41% reported willingness to receive the COVID-19 vaccines [93]. The most frequently (82%) cited concern was vaccine safety for their pregnancy. Non-Hispanic Black and Hispanic women were less likely to accept a COVID-19 vaccine compared to non-Hispanic White women (adjusted odds ratios [aOR] 0.4, 95% CI, 0.2–0.6 for both) [93]. In a study by Germann et al. on 456 individuals (93% pregnant, 7% postpartum) that examined the association of initial COVID-19 vaccine hesitancy with subsequent vaccination, the authors found that COVID-19 vaccine hesitancy persisted during the peripartum period [94]. They also observed that women who were older, parous, employed, with higher education level were more likely to accept a COVID-19 vaccine [94]. Women who identified as non-Hispanic Black, were Medicaid beneficiaries, and were still pregnant at follow-up were less likely to be vaccinated [94]. Taken together, findings from these studies highlight the racial and ethnic disparities in COVID-19 vaccine confidence and uptake among pregnant minority women in the US.

## 11. Racial Disparities Experienced by Pregnant Minority Women Due to the COVID-19 Pandemic and the Potential Impact on Pregnancy Care

In 2020, the COVID-19 pandemic disrupted prenatal care and birth plans for most pregnant women in the US [95]. The pandemic greatly disrupted obstetric care practices as new protocols were developed to keep mothers, babies, and staff safe [95]. Changes were made to prenatal care and birth plans such as shifts from in-person visits and increased cesarean delivery [96]. Clinicians had fewer opportunities to engage with pregnant women, especially pregnant minority women, who are at higher risk of severe COVID-19 disease as well as adverse maternal and neonatal outcomes. Clinicians also had fewer opportunities to counsel these pregnant women about preventive measures and the benefits of vaccination. Barriers for minority mothers to access COVID-19 vaccines require further investigation. Minority women are disproportionately impacted by disruptions in pregnancy care and routine vaccinations. Misinformation linking COVID-19 vaccines to infertility has increased COVID-19 vaccine hesitancy among minority mothers [97]. Minority mothers experience disproportionate community-level factors, such as increased housing density, differential ability to social distance due to occupation, and underlying medical conditions that put them at greater risk of severe COVID-19-associated complications [98]. Minority communities in the US commonly experience higher rates of pollution and poverty, lower socioeconomic status, and an increased risk of violence that can affect the mental and physical health of pregnant minority women [99]. Domestic violence towards women also appeared to spike during the COVID-19 pandemic [100]. Women experienced higher rates of unemployment during the pandemic than men, and working mothers were overwhelmed with increased childcare demands [100]. Residential segregation has been associated with high COVID-19 risk, possibly due to the abundance of multigenerational households in minority communities that may result in overcrowded living spaces unsuitable for quarantining [101,102]. African American and Hispanic Latinx communities are disproportionately burdened by the COVID-19 pandemic; however, they face difficulty accessing testing facilities that are often concentrated in wealthier neighborhoods [103]. African American and Hispanic Latinx communities are more likely to encounter maternity care deserts due to the minimal availability of obstetricians in their communities and the lack of insurance coverage [104,105]. Closure of hospitals serving low-income patients, limited options for public transportation, closure of obstetric wards, and poor access to primary care providers can all contribute to an increased risk of maternal exposure to COVID-19 among pregnant minority women in the US [106].

## 12. Strategies to Improve COVID-19 Vaccine Confidence and Uptake among Pregnant Minority Women in the US

A study by Razzaghi et al., including 135,968 pregnant women, analyzed vaccination coverage among US minorities, showing the following percentages: 24.7% for non-Hispanic Asians, 19.7% for non-Hispanic White women, 11.9% for Hispanic women and 6% for non-Hispanic Blacks; those results suggest that there is a need to improve outreach and engagement among pregnant minority women who might be at higher risk for severe health outcomes because of COVID-19. [107] Twelve strategies have been suggested by the CDC to increase vaccine confidence and demand in communities that include vaccine ambassadors, financial incentives, effective messages delivered by trusted messengers medical provider standardization, workplace vaccinations medical reminders, motivational interviewing, school-located vaccination programs, provision of recommendations, and combating misinformation [108].

According to the SAGE Working Group on Vaccine Hesitancy, hesitancy is complex and context-specific, varying across time, place and vaccines [109]. There may also be elements of vaccine hesitancy that are specific to race/ethnicity. Institutional racism is also an ever-present driver of vaccine distrust among Blacks in the US [110]. Historical injustices such as the Tuskegee syphilis study and unethical use of cancer cells from Henrietta Lacks are part of the foundation for distrust of the health care system among Blacks in the US [111,112,113]. There is a paucity of vaccine hesitancy data specific to pregnant minority populations in the US; however, public health intervention strategies to improve vaccine confidence and uptake have been based on the “knowledge deficit” approach, suggesting that improving vaccine-hesitant populations’ knowledge about the vaccines would improve vaccine acceptance [114]. Studies suggest that decision-making when it comes to vaccine acceptance is more complex and may include cultural, emotional, social, spiritual and political ideologies as well as cognitive factors that can drive vaccine acceptance [115,116,117]. In addition, many strategies have been examined to address vaccine hesitancy and improve vaccine uptake. There is no strong recommendation that a standard strategy will work better or for all populations experiencing vaccine hesitancy [118]. The foundation of most strategies to address vaccine hesitancy is to inform and educate [119,120]. However, this approach alone has provided inconsistent improvements in vaccine uptake [121,122]. In a study by Odone et al., they examined interventions that employed internet and social media as strategies to improve vaccine uptake and found that accessing vaccination campaign websites, text messaging, using patient-held web-based portals and computerized reminders may increase vaccine uptake [123]. However, it was unclear if social networks, email communication and smartphone applications helped to improve vaccine uptake [123]. Interventions recommended by the Community Prevention Services Task Force, that have been shown to be effective in improving vaccine uptake, include provisions for transportation or child care, food vouchers, gift cards, lottery prizes, and baby products [124]. This strategy can also be a part of a multicomponent strategy to address issues involving lack of vaccine access, appointment reminder and recall opportunities, health provider engagement for vaccines, and scheduled vaccinations for daycare and school entry [124]. Studies by Leask et al. suggest that when examining the literature two major drivers of vaccine hesitancy emerge, the influence of social norms and the interactions with healthcare providers [125]. This assessment would have significant implications for minority populations managing the constant influx of vaccine misinformation along with longstanding distrust of the healthcare providers. Peer-to-peer communication to improve vaccine confidence and uptake has also been considered and will require further study [126]. Taken together, strategies to address vaccine hesitancy require a planning framework that considers the WHO Guide to Tailoring Immunization Programs. The strategy should include multiple interventions that are data driven, that consider the hesitant population, specific drivers of vaccine hesitancy in the population, barriers to vaccine confidence, and defined means of evaluating the interventions.

Strategies to improve vaccine uptake among hesitant pregnant minority women that are likely to be successful would include an educational and information component, wrap-around services such as child care, food vouchers, gift cards, and baby products, health provider engagement, as well as peer-to-peer communication components. These recommendations in support of improving vaccine confidence and uptake among pregnant minority women are based on some of previous studies mentioned above. However, to accomplish this at scale requires a planning framework that considers the WHO Guide to Tailoring Immunization Programs. A major limitation of these previous studies is the non-significant inclusion of minority women, which will require additional studies.

Combining some of these strategies could be effective in improving vaccine confidence and uptake among pregnant minority women. Minority communities in the US have been heavily influenced by misinformation and fear of vaccine side effects for themselves and their children. This has resulted in vaccine hesitancy among pregnant minority women. We must engage expectant minority mothers early in their pregnancy to provide awareness, education, training, and access to routine vaccinations. The focus should be on implementing strategies that aim to improve vaccine confidence among expectant minority mothers in order to increase vaccine uptake so that they can achieve their health and wellness milestones. This approach will also promote racial equity in pregnancy care. A vaccine ambassador program that includes minority mothers from communities with a history of poor vaccine confidence and vaccine uptake (influenza, Tdap, and COVID-19 vaccines) can help address the lack of vaccine confidence in these communities. There is also a need to educate minority mothers about the importance of vaccination via culturally competent educational training sessions delivered by a trained staff that includes expectant mothers and mothers who have given birth. The training session can be performed in person or virtually and made accessible on a variety of social media platforms. Finally, a partnership between health departments and mobile vaccination programs designed to deliver vaccines (influenza, Tdap, COVID-19) to these communities is also crucial. Table 1 shows COVID-19 vaccine-related challenges and potential solutions among pregnant minority women in the US (Table 1).

## 13. Conclusions

The COVID pandemic has greatly impacted routine clinical visits and scheduled vaccinations for underserved pregnant minority women in the US. Direct and indirect barriers to pregnancy care for minority mothers have increased their risk of catching COVID-19 and impacted their access to COVID-19 vaccines. Minority mothers in the US were already disproportionately affected by maternal morbidity and mortality prior to the COVID-19 pandemic, and these racial disparities have been exacerbated during the ongoing pandemic. Inequity in pregnancy care and COVID-19 vaccine access should be carefully monitored in order to improve maternal and neonatal clinical outcomes among minority mothers during the COVID-19 pandemic and beyond.

## Figures and Tables

**Figure 1 vaccines-10-02122-f001:**
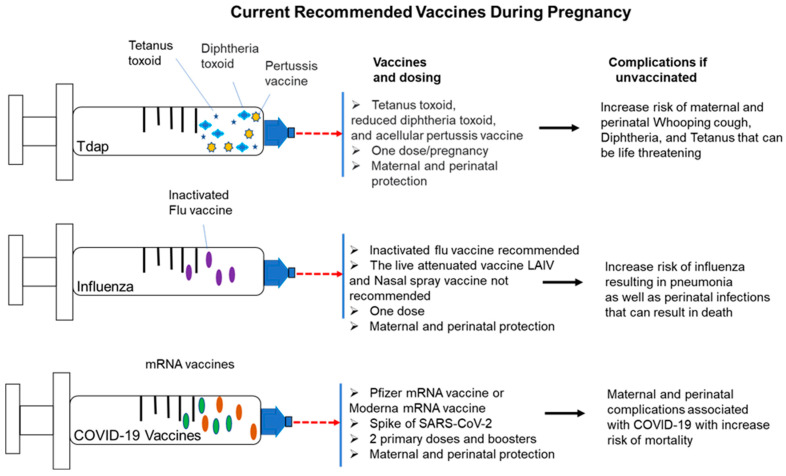
Currently recommended vaccines for pregnant women in the US.

**Figure 2 vaccines-10-02122-f002:**
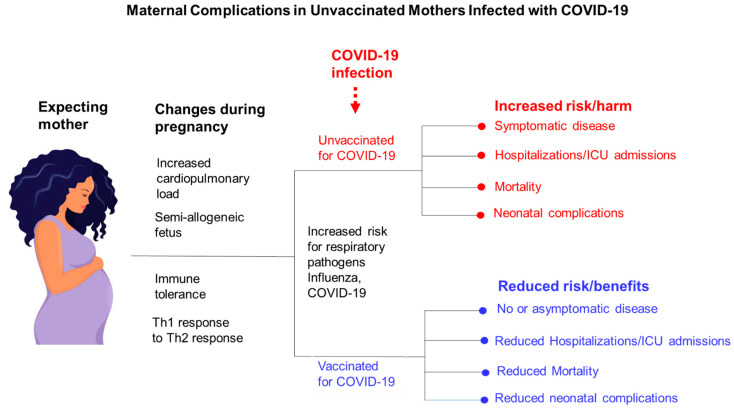
Maternal complications in unvaccinated mothers infected with COVID-19.

**Figure 3 vaccines-10-02122-f003:**
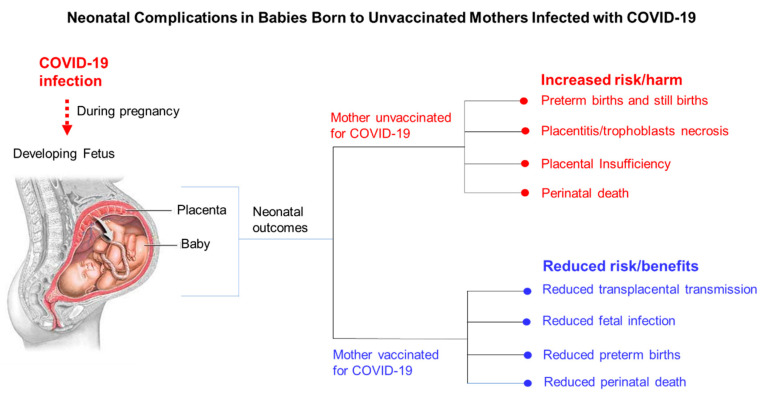
Neonatal complications observed in babies born to infected mothers who were unvaccinated for COVID-19.

**Table 1 vaccines-10-02122-t001:** Vaccine-related challenges and potential solutions among pregnant minority women.

Challenges	Solutions
Safety concerns and side effects for themselves and for their babies	Peer to peer communications to improve vaccine confidence and uptake
Distrust of medical providers and the government	Town hall meetings with pregnant minority women and medical providers of vaccines of the same race and ethnicity
Misinformation about the COVID-19 vaccine effects on fertility	Open discussions on social media platforms with medical providers and pregnant women to discuss vaccine safety regarding fertility
Unaware of the benefits of being vaccinated for COVID-19 during pregnancy	Community engagement health forums with pregnant women and OBGYN medical providers
Fear due to lack of research on the vaccines and its potential harm specific to minority communities	Community based focus groups with vaccinated and unvaccinated pregnant women that includes OBGYN medical providers providing culturally competent information

## Data Availability

This manuscript did not report any laboratory-based data.

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
