# Peer review of "The COVID-19 Vaccine and Pregnant Minority Women in the US: Implications for Improving Vaccine Confidence and Uptake"

_vaccines, 2022, doi:10.3390/vaccines10122122_

Round 1

Reviewer 1 Report

This paper is a very useful contribution as relates to pregnancy care and covid-19 vaccine access for underserved pregnant minority women in the US. 

I suggest to change "likely associated" into "likely to be associated" on line 23 

Author Response

November 16, 2022

Editor and Chief
Journal Vaccines

Manuscript ID vaccines-2010421

Dear Editor,

My responses to reviewers’ comments regarding manuscript ID vaccines-2010421 entitled “The COVID -19 Vaccine and Pregnant Minority Women in the US: Implications for Improving Vaccine Confidence and Uptake are enclosed.

Thank you for giving me the opportunity to resubmit this manuscript to your journal for publication.

Kind regards,

Donald J. Alcendor, Ph.D.

Associate Professor

Meharry Medical College

Center for AIDS Health Disparities Research

& Department of Microbiology and Immunology

& Obstetrics and Gynecology

Hubbard Hospital 5th Floor Rm. 5025

1005 Dr. D.B. Todd Jr. Blvd.

Nashville, TN 37208

Phone: 615-327-6449

Fax: 615-327-6929

Associate Professor Adjunct

Department of Pathology, Microbiology and Immunology

Vanderbilt University Medical Center

Reviewer 1

Comments and Suggestions for Authors

This paper is a very useful contribution as relates to pregnancy care and covid-19 vaccine access for underserved pregnant minority women in the US.

I suggest to change "likely associated" into "likely to be associated" on line 23

Author’s response:  We agree with the reviewer and the terms "likely associated" was changed to "likely to be associated" on line 23.  This change appears in blue text in revised manuscript.

Reviewer 2 Report

The present paper focuses on a very important and underestimated topical issue, as many periodic reports on vaccine safety do not specify which adverse reactions are most frequently observed in pregnant women.

I really appreciated the introduction section and the historical excursus on antiviral and antibacterial vaccinations in pregnancy. The section on the exclusion of pregnant women from clinical trials is also well-oriented on the problem; in my opinion, however, a brief speech could be implemented on the impossibility of the competent ethics committee to ascertain the best interest of the fetus, which is incapable of expressing its consent. The rest of the paper is complete and understandable, with oriented and logical conclusions including possible solutions for the future.

However, I have noticed some formal errors to double-check:

·         With reference to the contents of lines 135-137, it is evident that the sentence contained in lines 149-151 is a repetition. Looking at the relative citations (of which I recommend looking at the format as it is different between the two) it emerges that the (identical) data come from two different periods of 2021. Recheck the data and possibly eliminate the repetition.

·         At line 195 the acronym Tdap is introduced, which is described in the extended form only to line 204 (trivalent tetanus toxoid, reduced diphtheria toxoid, and acellular pertussis). Correct the error.

·         The concepts contained between lines 275-281, also repeated here, could be reformulated to be more fluent.

·         Does the study by Chinn et al cited between lines 289-292 refer to a cohort of pregnant women? Reading the original article it is clear that the sentence contained in the "Materials and Methods" section of the original article has been copied, without specifying whether the women were all pregnant. Specify the data.

·         Sentence at line 277 is referred to no. 77; does it is the correct reference? The study by Chinn et al is numbered at position 78 among the citations. Check and if necessary correct the error.

·         The sentence in lines 342-343 is erroneously repeated with respect to the above caption (Fig. 3). Correct the error.

·         The sentence in lines 422-427 referring to the study by Razzaghi et al is unclear from a syntactic point of view. I would suggest rephrasing as follows: “A study by Razzaghi et al. including 135,968 pregnant women analyzed vaccination coverage among US minorities, showing the following percentages: 24.7% for non-Hispanic Asians, 19.7% for non-Hispanic White women, 11.9% for Hispanic women and 6% for non-Hispanic Blacks; those results suggest that there is a need to improve outreach and engagement among pregnant minority women who might be at higher risk for severe health outcomes because of COVID-19”.

Furthermore, in my opinion some elements could be implemented:

·         brief statistical-demographic overview on the percentage representation of the minorities described with respect to the total population in the US since without this data it is difficult to understand the meaning of percentage data on vaccine hesitancy or complications from COVID-19 (eg lines 288-289);

·         references to vaccination surveillance data from other countries. In this regard, a retrospective assessment of adverse events in Italy was carried out in the paper: “Fazio ND, Delogu G, Bertozzi G, Fineschi V, Frati P. SARS-CoV2 Vaccination Adverse Events Trend in Italy: A Retrospective Interpretation of the Last Year (December 2020-September 2021). Vaccines (Basel). 2022;10(2):216” which might be useful to mention.

Author Response

November 16, 2022

Editor and Chief
Journal Vaccines

Manuscript ID vaccines-2010421

Dear Editor,

My responses to reviewers’ comments regarding manuscript ID vaccines-2010421 entitled “The COVID -19 Vaccine and Pregnant Minority Women in the US: Implications for Improving Vaccine Confidence and Uptake are enclosed.

Thank you for giving me the opportunity to resubmit this manuscript to your journal for publication.

Kind regards,

Donald J. Alcendor, Ph.D.

Associate Professor

Meharry Medical College

Center for AIDS Health Disparities Research

& Department of Microbiology and Immunology

& Obstetrics and Gynecology

Hubbard Hospital 5th Floor Rm. 5025

1005 Dr. D.B. Todd Jr. Blvd.

Nashville, TN 37208

Phone: 615-327-6449

Fax: 615-327-6929

Associate Professor Adjunct

Department of Pathology, Microbiology and Immunology

Vanderbilt University Medical Center

Reviewer #2

Comments and Suggestions for Authors

The present paper focuses on a very important and underestimated topical issue, as many periodic reports on vaccine safety do not specify which adverse reactions are most frequently observed in pregnant women.

I really appreciated the introduction section and the historical excursus on antiviral and antibacterial vaccinations in pregnancy. The section on the exclusion of pregnant women from clinical trials is also well-oriented on the problem; in my opinion, however, a brief speech could be implemented on the impossibility of the competent ethics committee to ascertain the best interest of the fetus, which is incapable of expressing its consent. The rest of the paper is complete and understandable, with oriented and logical conclusions including possible solutions for the future.

However, I have noticed some formal errors to double-check:

With reference to the contents of lines 135-137, it is evident that the sentence contained in lines 149-151 is a repetition. Looking at the relative citations (of which I recommend looking at the format as it is different between the two) it emerges that the (identical) data come from two different periods of 2021. Recheck the data and possibly eliminate the repetition. Authors’ response: We agree with the reviewer and line 149-151 have been deleted from the revised manuscript.

At line 195 the acronym Tdap is introduced, which is described in the extended form only to line 204 (trivalent tetanus toxoid, reduced diphtheria toxoid, and acellular pertussis). Correct the error.  Authors’ response: We agree with the reviewer and have spelled out the acronym at line 95 instead of line 204.  This change appears in blue text in the revised manuscript at line 195 and line 204.

The concepts contained between lines 275-281, also repeated here, could be reformulated to be more fluent.  Authors’ response: We agree with the reviewer and have deleted lines 278 to 281 in the revised manuscript.

Does the study by Chinn et al cited between lines 289-292 refer to a cohort of pregnant women? Reading the original article it is clear that the sentence contained in the "Materials and Methods" section of the original article has been copied, without specifying whether the women were all pregnant. Specify the data.  Authors’ response: The manuscript states that all women underwent childbirth with vs without COVID-19 between March 1, 2020, and February 28, 2021, at 499 US academic medical centers or community affiliates.  All women were pregnant in this study (all women underwent childbirth).

Sentence at line 277 is referred to no. 77; does it is the correct reference? The study by Chinn et al is numbered at position 78 among the citations. Check and if necessary correct the error.  Authors’ response: We agree with the reviewer and the reference number has been correctly changed to #78 and not #77.  This change appears in blue text of the revised manuscript.

The sentence in lines 342-343 is erroneously repeated with respect to the above caption (Fig. 3). Correct the error. Authors’ response: We agree with the reviewer and the repeated lines have been deleted at lines 342 and 343 in the revised manuscript.

  • The sentence in lines 422-427 referring to the study by Razzaghi et al is unclear from a syntactic point of view. I would suggest rephrasing as follows: “A study by Razzaghi et al. including 135,968 pregnant women analyzed vaccination coverage among US minorities, showing the following percentages: 24.7% for non-HispanicAsians, 19.7% for non-Hispanic White women, 11.9% for Hispanic women and 6%for non-Hispanic Blacks; those results suggest that there is a need to improve outreach and engagement among pregnant minority women who might be at higher risk for severe health outcomes because of COVID-19”.

Authors’ response: We agree with the reviewer and have changed the text accordingly as recommended by the reviewer and we want to thank the reviewer for this suggestion to improve the readability of the manuscript.  The revisions are shown in blue text in the revised manuscript.

Furthermore, in my opinion some elements could be implemented:  brief statistical-demographic overview on the percentage representation of the minorities described with respect to the total population in the US since without this data it is difficult to understand the meaning of percentage data on vaccine hesitancy or complications from COVID-19 (eg lines 288-289); Authors’ response: We agree with the reviewer and have added additional information in reference to the reviewers’ concerns.  The added information for stats in the population appears in blue text in the revised manuscript

references to vaccination surveillance data from other countries. In this regard, a retrospective assessment of adverse events in Italy was carried out in the paper: “Fazio ND, Delogu G, Bertozzi G, Fineschi V, Frati P. SARS-CoV2 Vaccination Adverse Events Trend in Italy: A Retrospective Interpretation of the Last Year (December 2020-September 2021). Vaccines (Basel). 2022;10(2):216” which might be useful to mention.  Authors’ response: We agree with the reviewer and have added information and the reference suggested by the reviewer to line 335 to 338 of the revised manuscript and the reference has been added accordingly to the reference selection as reference #87.

Reviewer 3 Report

1. I failed to learn what is the type of this review article. Without a clear methodology, the scientific value of review articles is very limited.

2. There are too many themes covered in this article which makes it too complicated to be comprehended.

3. Pregnant women's experiences of non-COVID-19 vaccines (e.g. flu, Tdap) did not help your narrative significantly. You can consider shortening this part.

4. I imagine that the core question in this article is the suboptimal uptake of the COVID-19 vaccine by minorities in the US; therefore, the manuscript needs to be structured around this topic.
You need to follow a theoretical framework to analyze the potential drivers of vaccine hesitancy, such as the 3C model of WHO-SAGE.
https://www.sciencedirect.com/science/article/pii/S0264410X15005009

Author Response

November 16, 2022

Editor and Chief
Journal Vaccines

Manuscript ID vaccines-2010421

Dear Editor,

My responses to reviewers’ comments regarding manuscript ID vaccines-2010421 entitled “The COVID -19 Vaccine and Pregnant Minority Women in the US: Implications for Improving Vaccine Confidence and Uptake are enclosed.

Thank you for giving me the opportunity to resubmit this manuscript to your journal for publication.

Kind regards,

Donald J. Alcendor, Ph.D.

Associate Professor

Meharry Medical College

Center for AIDS Health Disparities Research

& Department of Microbiology and Immunology

& Obstetrics and Gynecology

Hubbard Hospital 5th Floor Rm. 5025

1005 Dr. D.B. Todd Jr. Blvd.

Nashville, TN 37208

Phone: 615-327-6449

Fax: 615-327-6929

Associate Professor Adjunct

Department of Pathology, Microbiology and Immunology

Vanderbilt University Medical Center

Reviewer #3

Comments and Suggestions for Authors

  1. I failed to learn what is the type of this review article. Without a clear methodology, the scientific value of review articles is very limited. Authors’ response: The review article examines the impact of COVID-19 on minority women in the US that has influenced their attitudes about getting the recommended COVID-19 and how COVID-19 vaccine hesitancy has greatly impacted minority women in getting the other two ACOG recommended vaccines during pregnancy. The implication in the review is that COVID-19 hesitancy has created barriers for pregnant minority women in the US to access their regularly scheduled vaccines. Even more, it sheds light on pregnant women and their increase vulnerability to the most severe complications of COVID-19 if infected and are unvaccinated. 
  2. There are too many themes covered in this article which makes it too complicated to be comprehended. Authors’ response: The Tdap and influenza themes are together here with the COVID-19 vaccines because pregnant women since the pandemic have fallen behind on uptake of all three ACOG pregnancy recommended vaccines (influenza, Tdap, and COVID-19) which makes them and babies more susceptible to infection, disease and death if infected and unvaccinated.  This is especially significant among minority women in the US compared non-Hispanic White women.  Poor uptake of the COVID-19 vaccine as well as Tdap and influenza vaccines will disproportionately impact minority women resulting asymptomatic disease, hospitalizations and death associated with these preventable infections if unvaccinated.
  3. Pregnant women's experiences of non-COVID-19 vaccines (e.g. flu, Tdap) did not help your narrative significantly. You can consider shortening this part. Authors’ response: The Tdap and influenza themes are together here with the COVID-19 vaccines because pregnant women since the pandemic have fallen behind on uptake of all three ACOG pregnancy recommended vaccines (influenza, Tdap, and COVID-19) which makes them and babies more susceptible to infection, disease and death if infected and unvaccinated.  This is especially significant among minority women in the US compared non-Hispanic White women.  Poor uptake of the COVID-19 vaccine as well as Tdap and influenza vaccines will disproportionately impact minority women resulting asymptomatic disease, hospitalizations and death associated with these preventable infections if unvaccinated.
  4. I imagine that the core question in this article is the suboptimal uptake of the COVID-19 vaccine by minorities in the US; therefore, the manuscript needs to be structured around this topic.
    You need to follow a theoretical framework to analyze the potential drivers of vaccine hesitancy, such as the 3C model of WHO-SAGE.
    https://www.sciencedirect.com/science/article/pii/S0264410X15005009 Authors’ response: We disagree with the reviewer in that we have highlighted the potential drivers of COVID-19 vaccine hesitancy based on previous studies in this review. This is not a survey data driven manuscript designed to screen populations for factors driving vaccine hesitancy.  This however could be the subject of a future manuscript.

Round 2

Reviewer 3 Report

Dear authors,

Thank you for your esteemed efforts in responding to my previous points. Probably, there is a gap between our ways of thinking.

In your title, you stated that your article provides "implications" for minority women's vaccination uptake. This is a great and interesting question by the way, but how should the answer be acquired?. This is the question.

I assumed that you wanted to move from the UNKOWN to the KNOWN = from the QUESTION to the ANSWER, and per my humble experience, this journey requires clear, reproducible, and unbiased methodology, e.g. systematic review, scoping review, umbrella review, etc.

For example, in "scoping reviews", after we get the relevant results, we usually perform thematic analysis to describe key themes/concepts in the current scientific literature. In your article, you did the opposite thing, that's why I was confused and still not convinced why you have chosen to discuss these particular themes (Line 27 - 33).

The current manuscript can not provide any implications/recommendations for policymakers or clinical practitioners because it is far from being evidence-based work. However, it may serve well as educational material for students.

Alternatively, you can turn it into an "opinion" article, as it would make more sense.

P.S. I really appreciate your efforts in writing this massive amount of information and scanning this portion of the literature, and I am still trying to provide constructive critique for the work.

Sincerely,

Author Response

November 22, 2022

Editor and Chief
Journal Vaccines

Manuscript ID vaccines-2010421

Dear Editor,

My responses to reviewer #3 comments regarding manuscript ID vaccines-2010421 entitled “The COVID -19 Vaccine and Pregnant Minority Women in the US: Implications for Improving Vaccine Confidence and Uptake are enclosed.

Thank you for giving me the opportunity to resubmit this manuscript to your journal for publication.

Kind regards,

Donald J. Alcendor, Ph.D.

Associate Professor

Meharry Medical College

Center for AIDS Health Disparities Research

& Department of Microbiology and Immunology

& Obstetrics and Gynecology

Hubbard Hospital 5th Floor Rm. 5025

1005 Dr. D.B. Todd Jr. Blvd.

Nashville, TN 37208

Phone: 615-327-6449

Fax: 615-327-6929

Associate Professor Adjunct

Department of Pathology, Microbiology and Immunology

Vanderbilt University Medical Center

Reviewer #3

Comments and Suggestions for Authors

Thank you for your esteemed efforts in responding to my previous points. Probably, there is a gap between our ways of thinking.

In your title, you stated that your article provides "implications" for minority women's vaccination uptake. This is a great and interesting question by the way, but how should the answer be acquired?. This is the question.

I assumed that you wanted to move from the UNKOWN to the KNOWN = from the QUESTION to the ANSWER, and per my humble experience, this journey requires clear, reproducible, and unbiased methodology, e.g. systematic review, scoping review, umbrella review, etc.

For example, in "scoping reviews", after we get the relevant results, we usually perform thematic analysis to describe key themes/concepts in the current scientific literature. In your article, you did the opposite thing, that's why I was confused and still not convinced why you have chosen to discuss these particular themes (Line 27 - 33).

The current manuscript can not provide any implications/recommendations for policymakers or clinical practitioners because it is far from being evidence-based work. However, it may serve well as educational material for students.

Alternatively, you can turn it into an "opinion" article, as it would make more sense.

P.S. I really appreciate your efforts in writing this massive amount of information and scanning this portion of the literature, and I am still trying to provide constructive critique for the work.

Sincerely,

Authors’ response

We somewhat agree with the reviewer and have added additional information to broadly address the reviewer concerns that appears in red text in the revised manuscript. There are also 18 additional references to support the added information.  The references appear in red text in the Reference section of the revised manuscript.

Round 3

Reviewer 3 Report

Dear author(s),

Thank you for your esteemed efforts. I appreciate your efforts in writing this long article; however, I do not believe that your methodology is reproducible or objective. Therefore, I am still sceptical about the usability of this manuscript.

Please change your article type from "Review" to "Opinion".

Sincerely,

Author Response

November 25, 2022

Editor and Chief
Journal Vaccines

Manuscript ID vaccines-2010421

Dear Editor,

My responses to reviewer #3 comments regarding manuscript ID vaccines-2010421 entitled “The COVID -19 Vaccine and Pregnant Minority Women in the US: Implications for Improving Vaccine Confidence and Uptake are enclosed.  In all capacity I feel that I have addressed the reviewers’ comments with additional information that is supported by 18 additional references that appears in red text in the revised manuscript.  I do not think additional information added to manuscript will satisfy reviewer #3.  There is limited data in the literature regarding specific strategies to improve vaccine confidence and uptake among pregnant minority women in the US. Therefore, I am making an appeal to the academic editor.

Thank you for giving me the opportunity to resubmit this manuscript to your journal for publication.

Kind regards,

Donald J. Alcendor, Ph.D.

Associate Professor

Meharry Medical College

Center for AIDS Health Disparities Research

& Department of Microbiology and Immunology

& Obstetrics and Gynecology

Hubbard Hospital 5th Floor Rm. 5025

1005 Dr. D.B. Todd Jr. Blvd.

Nashville, TN 37208

Phone: 615-327-6449

Fax: 615-327-6929

Associate Professor Adjunct

Department of Pathology, Microbiology and Immunology

Vanderbilt University Medical Center

Reviewer #3

Comments and Suggestions for Authors

Thank you for your esteemed efforts in responding to my previous points. Probably, there is a gap between our ways of thinking.

In your title, you stated that your article provides "implications" for minority women's vaccination uptake. This is a great and interesting question by the way, but how should the answer be acquired? This is the question.

I assumed that you wanted to move from the UNKOWN to the KNOWN = from the QUESTION to the ANSWER, and per my humble experience, this journey requires clear, reproducible, and unbiased methodology, e.g. systematic review, scoping review, umbrella review, etc.

For example, in "scoping reviews", after we get the relevant results, we usually perform thematic analysis to describe key themes/concepts in the current scientific literature. In your article, you did the opposite thing, that's why I was confused and still not convinced why you have chosen to discuss these particular themes (Line 27 - 33).

The current manuscript can not provide any implications/recommendations for policymakers or clinical practitioners because it is far from being evidence-based work. However, it may serve well as educational material for students.

Alternatively, you can turn it into an "opinion" article, as it would make more sense.

P.S. I really appreciate your efforts in writing this massive amount of information and scanning this portion of the literature, and I am still trying to provide constructive critique for the work.

Sincerely,

Authors’ response

We somewhat agree with the reviewer and have added additional information to broadly address the reviewer concerns that appears in red text in the revised manuscript. There are also 18 additional references to support the added information.  The references appear in red text in the Reference section of the revised manuscript.